

# The geothermal gradient from mesophilic to thermophilic temperatures shapes microbial diversity and processes in natural gas-bearing sedimentary aquifers

Taiki Katayama[1], Hideyoshi Yoshioka[1], Toshiro Yamanaka[2], Susumu Sakata[1] and Yasuaki Hanamura[3]

[1]Institute for Geo-Resources and Environment, Geological Survey of Japan (GSJ), National Institute of Advanced Industrial Science and Technology (AIST), Ibaraki 305-8567, Japan.
[2]School of Marine Resources and Environmental Sciences, Tokyo University of Marine Science and Technology, Tokyo 108-8477, Japan.
[3]JX Nippon Oil and Gas Exploration Corporation, Tokyo 100-8163, Japan.

*Correspondence to*: Taiki Katayama (katayama.t@aist.go.jp)

**Abstract.** The majority of Earth's prokaryotes live under the deep sedimentary biosphere. Geochemical processes driven by geothermal heating may play a crucial role in fueling deep subsurface microbial biomass and activities, yet their full breadth remains uncaptured. Here, we investigated the microbial community composition and metabolism in microbial natural gas-bearing aquifers at temperatures ranging from 35−80 °C, situated above nonmicrobial gas and oil-bearing sediments at temperatures exceeding 90 °C. Cultivation-based and molecular gene sequencing analyses, including radiotracer measurements, of formation water indicated variations in predominant methanogenic pathways across different temperature regimes of upper aquifers: high potential for hydrogenotrophic/methylotrophic, hydrogenotrophic and acetoclastic methanogenesis at depths with mesophilic, thermophilic and hyperthermophilic temperatures, respectively. The potential for acetoclastic methanogenesis correlated with elevated acetate concentrations with increasing depth, possibly due to the thermal decomposition of sedimentary organic matter. In addition to acetoclastic methanogenesis, in aquifers with hyperthermophilic temperatures, acetate is potentially utilized by microorganisms responsible for the dissimilatory reduction of sulfur compounds other than sulfate because of its high relative abundance at greater depths. The stable sulfur isotopic analysis of sulfur compounds in water and oil samples suggested that hydrogen sulfide generated through the thermal decomposition of sulfur compounds in oil migrates upward and is subsequently oxidized with iron oxides present in sediments, yielding elemental sulfur and thiosulfate. These compounds are consumed by sulfur-reducing microorganisms, possibly reflecting elevated microbial populations in aquifers with hyperthermophilic temperatures. These findings reveal previously overlooked geothermal heat-driven geochemical and microbiological processes involved in carbon and sulfur cycling in the deep sedimentary biosphere.



## 1 Introduction

Deep subsurface environments harbor a large fraction of the prokaryotes present on Earth (Mcmahon and Parnell, 2014; Magnabosco et al., 2018), exceeding 80% of the total prokaryotic biomass (Bar-On et al., 2018). Aquifers that form in sedimentary environments provide microorganisms with pore spaces, water and buried organic materials that serve as energy and carbon sources, sustaining metabolic activities (Mcmahon and Chapelle, 1991; Lovley and Chapelle, 1995; Fredrickson et al., 1997; Krumholz et al., 1997). In deep buried sediments, geothermal heating drives chemical reactions, including the

thermal decomposition of sedimentary organic matter, which may also generate substances essential for subsurface microbial activities (Bottrell et al., 2000; Parkes et al., 2014; Wellsbury et al., 1997). This biosphere–geosphere interaction (Parkes et al., 2014) in deep buried sediments is evident in petroleum reservoirs, as oil is generated from chemical transformation by geothermal heat and pressure, and the biodegradation of oil enhances microbial biomass (Bennett et al., 2013). Recently, such elevated microbial cells and activity were also found at the interface between deep buried sediments and underlying basement

rocks (Heuer et al., 2020).

The natural gas-bearing sedimentary aquifers targeted in this study are located beneath the Echigo Plain in the coastal area of the Japan Sea (Fig. 1a). In this gas field, gases are dissolved in the formation water (FW) and produced for commercial purposes by pumping gas-associated FW from the upper formations (Haizume and Nishiyama Formations, Fig. 1b). Additionally, due to the occurrence of deeply buried sedimentary rocks rich in organic matter (Suzuki et al., 1995) and a steep

geothermal gradient of approximately 5 °C per 100 m (Kato, 2018), oil and associated gases have also been deposited and commercially produced from the lower formation (the Shiiya Formation, Fig. 1b) (Fukano et al., 2023). The consequence of this high geothermal gradient is that the upper dissolved natural gas-bearing aquifers exhibit a wide temperature range, spanning approximately 35 to 80 °C (Fig. 1c), while temperatures in the lower oil deposits exceed 90 °C. This suggests a potential boundary between the biosphere (i.e., upper aquifers) and the geosphere (i.e., lower oil reservoirs), which has been

estimated at 80-90 °C in sedimentary environments (Head et al., 2003). Indeed, the chemical and isotopic compositions of natural gases (Kaneko and Igari, 2020) indicate that methane dissolved in upper aquifers is primarily of microbial origin, whereas that from lower oil deposits primarily originates from oil-associated thermogenic processes (Fig. S1) based on the classification by Milkov and Etiope (Milkov and Etiope, 2018). Therefore, the upper aquifers exhibit unique geochemical characteristics, not only encompassing a broad temperature spectrum ranging from mesophilic to hyperthermophilic conditions

for microorganisms but also extending to an unexplored interface between the biosphere and the geosphere associated with oil.



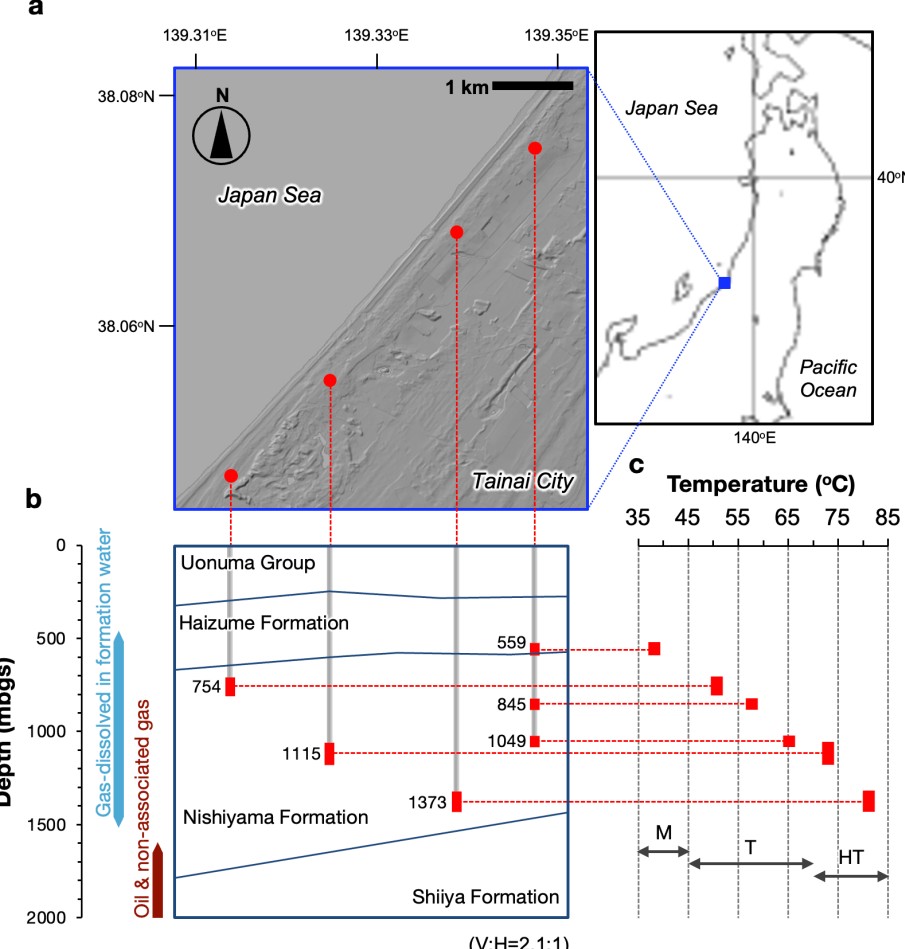

**Fig. 1.** Geological map of the study site (a) and relationship between the depths of the sampled wells (b) and the measured water temperatures (c) for microbiological analyses. The ratio of the vertical to horizontal scale of the cross section is 2.1:1 (b). The range of the well strainers is indicated by a red bar (b, c), and the average depth (mbgs) of each well strainer is also indicated (b). Because three wells are present at the same location, only one well is shown in panel b. Abbreviations: M, mesophilic range; T, thermophilic range; HT, hyperthermophilic range.

The objective of this study was to investigate how microbial communities and metabolic mechanisms participating in carbon and sulfur cycles are shaped by geochemical processes associated with both geothermal heating and oil occurrence. FW samples were collected from the upper aquifers with different reservoir layers with water temperatures ranging from approximately 35 to 80 °C and from the lower oil deposits. The combination of cultivation-based approaches, including radiotracer experiments, molecular sequencing microbiological analyses, and geochemical analyses, provided a



comprehensive understanding of microbial diversity, community compositions, and potential metabolism in the deep
sedimentary aquifer systems.

## 2 Materials and Methods

### 2.1 Site description and sample collection

The study site was located along the Japan Sea coast in Tainai city, Niigata Prefecture, Japan (Fig. 1). The reservoirs are
distributed in the Neogene and Quaternary formations of the Niigata sedimentary basin. The reservoirs of dissolved natural
gases are distributed within the Pliocene to Pleistocene Nishiyama and Pleistocene Haizume Formations and mainly consist of
sandstone or conglomerate (Kobayashi, 2000; Fukano et al., 2023). These sediments are interpreted as turbidite sediments
deposited under submarine fan to shelf conditions (Takano et al. 2001). Logging analysis of the gas production wells revealed
that each reservoir layer was separated by silt-mudstone, and the horizontal continuity was traced. Oil and associated gas are
deposited along anticlinal structures in the lower part of the Nishiyama Formation and the Pliocene Shiiya Formation (Fukano
et al. 2023). The reservoirs of oil deposits consist of sandstone in turbidite sediments deposited under submarine fan and shelf
conditions (Takano et al. 2001).

The FW samples subjected to microbiological analyses were collected from the separate reservoir layers in the Haizume
and Nishiyama Formations through 6 commercial production wells with depths ranging from approximately 500 to 1430 mbgs
(Fig. 1). Crude oil and associated FW samples were also collected from the Shiiya Formation through tanks separating oil, FW
and gas produced from multiple wells and used for microbial cell counting and geochemical analysis. The geological
information and analytical purpose of each sample are summarized in Table S1.

FW samples for methanogen activity measurements and cultivation were collected in sterilized glass bottles with butyl
rubber stoppers and screw caps. The bottles were purged with $N_2$ gas before and during sample collection and then filled with
FW to prevent the penetration of air. For the molecular analysis, 2-L FW samples were collected and filtered through a 0.2-
μm pore size Millipore Express Plus membrane filter (Millipore, Billerica, MA, USA) to harvest microbial cells, which were
subsequently stored at -20 °C. The samples used for total cell counts were fixed with formalin at a final concentration of 2%
(v/v) immediately after sampling and stored at 4 °C. Zinc acetate solution (2.0 M) was added to the water sample to fix the
hydrogen sulfide dissolved in the FW immediately after collection, and the solution was used for the measurement of the
hydrogen sulfide concentration.

### 2.2 Geochemical analysis

The inorganic chemical and total organic carbon contents in the FW samples were measured as described previously (Katayama
et al. 2015), and the analytical methods used for each compound are summarized in Table S2. The concentrations of formate,
acetate, propionate and butyrate were measured using a Prominence HPLC system (Shimadzu Co. Ltd., Kyoto, Japan)
equipped with an electrical conductivity detector. The concentration of the fixed hydrogen sulfide sample was measured
colorimetrically with a SmartSpecTM Plus spectrophotometer (Bio-Rad, CA, USA) using ferrous hydrogen sulfide reagents



(HACH, Loveland, CO, USA). Hydrocarbon components in FW samples from upper aquifers were extracted and concentrated with dichloromethane and qualitatively analyzed using gas chromatography–mass spectrometry (5973 GC–MS, Agilent Technologies, CA). The crude oil sample from the oil deposits was diluted with hexane and qualitatively analyzed via GC–MS. The concentration of phenols in the FW sample was measured via the 4-aminoantipyrine spectrophotometric method. The

concentrations of methanol, toluene and xylene in the FW sample were measured using a GC–MS (Agilent 5973, Agilent Technologies) instrument equipped with a headspace sampler (Agilent 7697A, Agilent Technologies).

   The stable oxygen and hydrogen isotopic compositions of the FW samples were measured via wavelength-scanned cavity ring-down spectroscopy (WS-CRDS) via a laser spectroscopy analyzer (Picarro L2120-i).

   Stable sulfur isotope analysis of sulfate and hydrogen sulfide dissolved in FW was performed as described previously

(Yamanaka et al., 2013; Katayama et al., 2019). For stable isotope analysis of elemental sulfur (dissolved in FW), hydrogen sulfide dissolved in FW samples was first removed by precipitation using zinc acetate solution, followed by filtration using filter paper to trap the generated zinc sulfide. Liquid–liquid extraction of elemental sulfur was performed from the filtrate with carbon disulfide followed by evaporation of carbon disulfide in a sand bath (~60 °C), and crystallized elemental sulfur was measured by an elemental analyzer/isotope ratio mass spectrometer (IsoPrime EA; GV Instruments, Manchester, UK). For

stable sulfur isotope analysis of sulfur compounds, sulfur compounds in oil were oxidized and decomposed in the high-pressure oxygen combustion unit Parr Bomb and converted to barium sulfate. The precipitate was wrapped in a tin capsule containing $V_2O_5$ and measured by an elemental analyzer/isotope ratio mass spectrometer.

## 2.3 Methanogenic activity measurement using a radiotracer

The methane production rates in the FW samples were measured using radiotracer experiments as described previously (Katayama et al. 2015). Briefly, a 20 ml water sample in a 50 ml serum vial sealed with a butyl rubber stopper and aluminum crimp was injected with either the radiotracers $^{14}C$-bicarbonate (10 ml, 199 kBq), [2-$^{14}C$]-acetate (10 ml, 81 kBq) or $^{14}C$-methanol. The plants were incubated under an atmosphere of $N_2/CO_2$ (80:20, v/v) for 14, 28 or 42 days. The activity measurements were conducted in triplicate for each cultivation period. The incubation temperatures were set close to the

measured water temperature of each sample (i.e., 559 mbgs, 37 °C; 754 mbgs, 50 °C; 845 mbgs, 55 °C; 1049 mbgs, 65 °C; 1115 mbgs, 75 °C; 1373 mbgs, 80 °C). The produced $^{14}CH_4$ was oxidized to $^{14}CO_2$ and measured using a Tri-Carb 3100TR liquid scintillation counter (Perkin Elmer). The $^{14}C$ activity at time zero was used as a control. The methane production rate was calculated using the equation $ap/(art)C$, where $ap$ and $ar$ are the activities of the product and added reactant, respectively; $t$ is the incubation period; and C is the *in situ* concentration of the reactant.


## 2.4 Direct cell counts

A formation water sample was filtered through a 0.2-μm-pore-size Isopore membrane filter (Millipore), stained for 10 min with SYBR Green solution (10 μg ml$^{-1}$) and observed under an epifluorescence microscope (BX51; Olympus, Tokyo, Japan).



## 2.5 Quantitative PCR

DNA was extracted from the membrane filter filtered through formation water samples using a PowerWater kit (MoBio Laboratories, CA, USA) according to the manufacturer's protocol. Quantitative PCR targeting bacterial and archaeal 16S rRNA genes and the *murA* gene (encoding a methyl-coenzyme M reductase alpha subunit) in formation water samples was performed as described previously (Katayama et al. 2016).


## 2.6 Sequencing of the 16S rRNA and mcrA genes

The 16S rRNA genes (including variable regions 4 and 5) were amplified and sequenced as described previously (Katayama et al. 2016). In brief, the primers Univ515F (5'-GTGYCAGCMGCCGCGGTA) and Univ926R (5'-CCGYCAATTCMTTTRAGTT) were used for PCR amplification. Amplicons were sequenced on an Illumina MiSeq platform

(Illumina, Inc., San Diego, CA, USA) using a 250 bp paired-end protocol at the J-Bio21 Center, Nippon Steel and Sumikin Eco-Tech Corporation (Tsukuba, Japan).

The *mcrA* genes were amplified, cloned and sequenced from the FW and methanogen culture samples (as described below) as described previously (Katayama et al. 2016) with the following modifications. The primers MLf and MLr (Luton et al. 2002) were used for PCR amplification, and clones were subjected to Sanger sequencing using Eurofin Sanger sequencing

services (Tokyo, Japan).

## 2.7 Sequence analysis

The 16S rRNA gene amplicon reads were analyzed using the Fastx toolkit (ver 0.0.14) and QIIME 2 (ver 2022.2). The quality-filtered sequences with an average length of 375 bp were clustered as amplicon sequence variants (ASVs) based on the DADA2

algorithm and classified using a Bayesian classifier based on the Silva taxonomy SSU Ref 138.1 dataset (Pruesse et al. 2007) with a confidence threshold of 80%.

Sanger sequences of the *mcrA* genes derived from the FW and culture samples were aligned using MAFFT version 7 (Katoh and Standley 2013). Nucleotide sequences with >85% sequence identity were treated as operational taxonomic units (OTUs) according to the cutoff values estimated by Yang et al. (Yang et al. 2014). For each OTU, the most abundant sequence

was selected as the representative sequence, which was translated to amino acids *in silico* and then searched against BLAST (http://blast.ncbi.nlm.nih.gov/Blast.cgi).

The 16S rRNA gene amplicon data were submitted to the DDBJ Sequence Read Archive database under accession number PRJDB16863. The GenBank/EMBL/DDBJ accession numbers for the *mcrA* gene sequences are LC783983-LC783991.

## 2.8 Cultivation of methanogens

The basal media used for the methanogenic cultures consisted of 10 mM $NH_4Cl$, 1 mM $KH_2PO_4$, 15 mM $MgCl_2 \cdot 6H_2O$, 350 mM NaCl, 1 mM $CaCl_2 \cdot 2H_2O$, 30 mM $NaHCO_3$, 1 ml l$^{-1}$ selenium and tungsten solution, 1 ml l$^{-1}$ trace element solution, 1 ml l$^{-1}$ vitamin solution, 1 ml l$^{-1}$ resazurin solution (1 mg ml$^{-1}$), and 0.1 mM titanium(III) citrate (as a reducing agent) (Katayama



et al. 2020). Using an anaerobic chamber, 40 ml of the FW sample was mixed with 20 ml of this saline mineral medium and
then dispensed into 110-mL serum vials. The vials were sealed with butyl rubber septa and aluminum crimps under an
atmosphere of $N_2/CO_2$ (80:20, v/v). The culture was supplemented with either $H_2/CO_2$ (80:20, v/v; 0.1 MPa) plus formate (20
mM) plus yeast extract (0.02%) (for culturing hydrogenotrophic methanogens), acetate (20 mM) (for culturing acetoclastic
methanogens) or methanol (10 mM) plus trimethylamine (10 mM) (for culturing methylotrophic methanogens). The incubation
temperatures were set at a value close to the measured water temperature of each sample (as described for the radiotracer
measurement methods). Methane production was measured using a GC equipped with a TCD. After methane production was
terminated, 4 mL of the sample cultures were harvested by centrifugation, and *mcrA* gene sequencing analysis was conducted
as described above.

## 3 Results

### 3.1 Geochemistry of the formation water

The geochemical properties of the FW samples are summarized in Table S3. The FW from the upper aquifers was characterized
by higher concentrations of bicarbonate and ammonium ions, comparable concentrations of chloride and sodium ions, and
almost no sulfate in comparison with present-day seawater. The depletion of electron acceptors for microbes, such as nitrate,
nitrite and sulfate, resulted in a low redox potential of FW lower than -180 mV.

The crude oil sample from the lower oil deposits contained n-alkanes with carbon numbers of 9-35, whereas such alkanes
were not detected in the FW samples from the upper aquifers (Fig. S2). On the other hand, volatile hydrocarbons such as
toluene and xylene were detected in the FW samples from the upper aquifers and were most likely derived from oil. In the
1049 and 1115 mbgs samples, elemental sulfur was also detected. The concentrations of phenols dissolved in the FW samples
increased from 1115 mbgs to greater depths (above >73 °C), whereas toluene or xylene did not (Table S3). Similar increasing
trends at greater depths were also observed for formate, acetate, propionate and total organic carbon. A steep increase in acetate
concentration at >70 °C has also previously been observed in deep subseafloor sediments (Heuer et al. 2020; Parkes et al.
2007), possibly due to the decomposition of sedimentary organic matter along with elevated geothermal heating (Parkes et al.
2007). Considering that phenols are prominent pyrolysis products of terrestrial kerogens according to previous studies (Larter
and Senftle 1985; van de Meent et al. 1980), it is conceivable that the presence of these compounds is also associated with the
thermal decomposition of sedimentary organic matter, similar to acetate.

### 3.2 Stable isotopic analysis of sulfur compounds

Fig. 2 illustrates the depth distribution of stable sulfur isotope ratios ~~of sulfur~~ in different sulfur species, namely, sulfide,
elemental sulfur and sulfate in FW and sulfur compounds in oil. Elemental sulfur exhibited the lowest $\delta^{34}S$ (-3.3 − -2.0‰),
followed by hydrogen sulfide (+1.5 − +9.4‰), sulfur compounds in oil (+8.6 − +9.6‰), and sulfate (+12.1 − +19.4‰).



Differences in $\delta^{34}$S values were not clearly observed between samples from upper aquifers and lower oil deposits for each sulfur species.

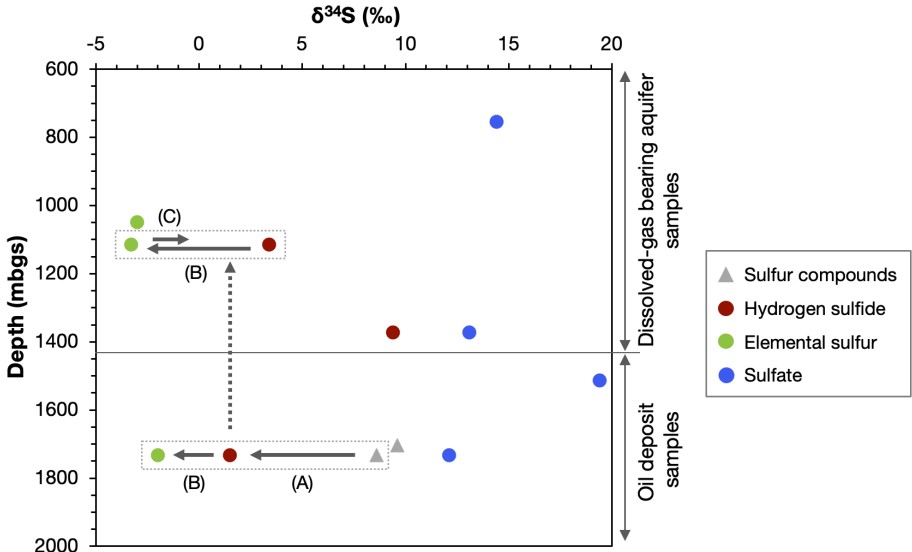

**Fig. 2.** Depth distribution of stable sulfur isotopic ratios in sulfur species in the formation water (FW, circle) and oil (triangle) samples. The arrows indicate the possible changes in the ratios due to isotopic fractionation associated with the following reactions: thermal decomposition of oil sulfur to hydrogen sulfide (A), oxidation of upward migrated hydrogen sulfide (broken arrow) with iron oxides to elemental sulfur (B) and biological dissimilatory reduction of elemental sulfur to hydrogen sulfide (C; for details, see the Discussion section).

**3.3 Hydrogen and oxygen isotope ratios of the formation water samples**

To assess the potential mixing of formation water (FW) from upper aquifers and lower oil deposits, isotopic values ($\delta^{18}$O and $\delta$D) of water samples collected from specific layers were determined. In the $\delta^{18}$O-$H_2$O vs. $\delta$D-$H_2$O diagram (Fig. S3), FW samples collected from aquifers above 1115 mbgs plot along the meteoric water line in this region (Kato and Kajiwara 1986; Waseda and Nakai 1983). In addition, the chlorine ion concentration of the FW sample at the uppermost depth of the 554 mbgs sample was lower than that of the present seawater (Table S3), suggesting the mixing of surface water in this sample. In samples below 1373 mbgs, there was a trend of increasing $\delta^{18}$O-$H_2$O concentration with increasing depth, possibly due to the isotopic exchange between formation water and sedimentary minerals at elevated temperatures (Clayton et al. 1966). The difference in water isotopic ratios, coupled with the presence of aquiclude silt-mudstone layers between (Takano et al. 2001), indicates no mixing of formation water between the upper aquifers and lower oil deposits.

**3.4 Potential for methanogenic activity**



The $^{14}C$ tracer experiments detected the potential activity of methanogenesis from $CO_2$, acetate and methanol (hydrogenotrophic, acetoclastic and methylotrophic pathways, respectively) in all 6 FW samples from the dissolved natural gas-bearing aquifers (Fig. 3a). The depth-related patterns of the activity rates clearly differed among these three methanogenic

pathways. Methylotrophic methanogenesis peaked at the shallowest sample (559 mbgs) and decreased by at least three orders of magnitude in deeper samples. The hydrogenotrophic methanogenic rates were greater in the three shallowest samples (< 845 mbgs) than in the deeper samples (> 1049 mbgs), and the maximum rates were observed in 845 mbgs. The activity rates of acetoclastic methanogenesis were at least two orders of magnitude greater in deeper samples (1115 and 1373 mbgs).

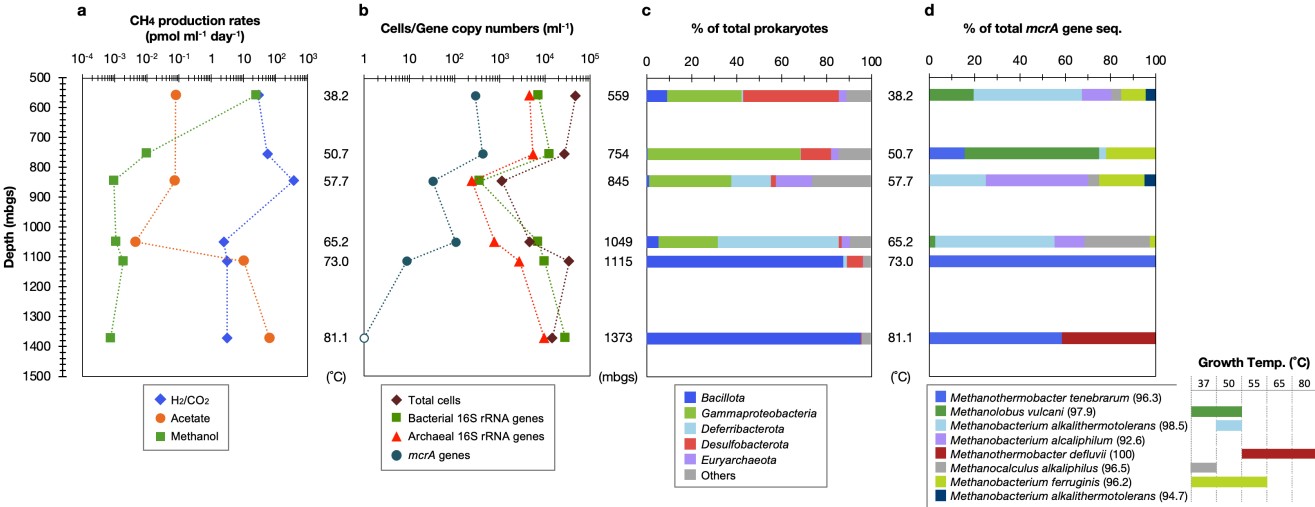

**Fig. 3.** Depth (temperature)-related changes in potential methanogenic activity (a), microbial population (b), and prokaryotic (c) and methanogen (d) community compositions based on the 16S rRNA (c) and *mcrA* (d) gene sequences in formation water samples from the upper aquifers. The growth temperatures (e) are expressed by the detection of *mcrA* gene sequences from both original formation water samples and methanogen cultures at different temperatures (for details, see Table S4).

**3.5 Enumeration of total microbial cells and of the 16S rRNA and mcrA genes**

The number of microbial cells in the FW samples ranged from $1.1\times10^3$ to $4.8\times10^4$ cells ml$^{-1}$ (Fig. 3b). The highest and lowest numbers were measured in the shallowest 559 and 845 mbgs samples, respectively. No microbial cells were observed in the oil-associated FW sample (1733 mbgs). Accordingly, molecular gene sequencing analysis was not conducted on this sample.

The populations of bacteria, archaea and methanogens were measured via quantitative real-time PCR (Fig. 3b). The
bacterial 16S rRNA gene copy numbers ranged from $10^2$-$10^4$ ml$^{-1}$, whereas those of archaea ranged from $10^2$-$10^3$ ml$^{-1}$. Bacterial and archaeal gene copy numbers were highest in the deepest sample (1373 mbgs). On the other hand, in this sample, the *mcrA* gene copy number was below the detection limits. The *mcrA* gene copy numbers tended to decrease with increasing depth.

**3.6 Microbial community compositions**





16S rRNA gene amplicon analysis was performed to examine the compositions of the prokaryotic and methanogenic communities. After quality filtering, the reads yielded 88,255-108,428 reads per sample. The major taxonomic groups (>10% of the total reads in at least one sample) were the phyla or classes *Bacillota* (formerly *Firmicutes*), *Gammaproteobacteria*, *Deferribacterota*, *Desulfobacterota* and *Euryarhcaeota* (Fig. 3c). The members of *Bacillota* were abundant in two deeper samples (1115 and 1373 mbgs), accounting for >85% of the total sequences. In the shallower two samples (559 and 754 mbgs),

*Gammaproteobacteria* and *Desulfobacterota* sequences were abundant, whereas *Gammaproteobacteria* and *Deferribacterota* sequences were abundant in the two intermediate depth samples (845 and 1049 mbgs). In the samples deeper than 1000 mbgs, only one ASV accounted for ≥50% of the total sequences in the sample. These genes were phylogenetically related to *Deferribacter desulfuricans* (100% sequence similarity, in the 1049 mbgs sample), *Thermacetogenium phaeum* (93% sequence similarity, in the 1115 mbgs sample) and *Thermanaeromonas toyohensis* (97% sequence similarity, in the 1373 mbgs sample)

(Table S1). These species can commonly utilize sulfur compounds, such as thiosulfate and elemental sulfur (but not sulfate), as electron acceptors and acetate as an electron donor (Takai et al., 2003; Hattori et al., 2000; Mori et al., 2002). Although acetate utilization by *Thermanaeromonas* has not been reported, all three genomes of this genus available at GTDB (including *T. toyohensis*) possess genes for the TCA cycle and the Wood–Ljungdahl pathway, suggesting that they may utilize acetate for the dissimilatory reduction of sulfur compounds.

The archaeal sequences accounted for 0.04-18.99% of the total sequences (Table S2). The majority (>3% of the total archaea) were assigned to *Halobacterota*, *Euryarchaeota* and *Thermoplasmatota*, whereas the minor sequences were *Ca*. Asgardarchaeota, *Ca*. Hadarchaeota and *Ca*. Nanoarcaheota. The sequences of hydrogenotrophic methanogens, such as *Methanobacterium* and *Methanocalculus*, were abundant in shallower depth samples (<1050 mbgs), whereas *Archaeoglobaceae* were abundant in deeper samples (1115 and 1373 mbgs). Members of the *Archaeoglobaceae* family are

known to reduce sulfur compounds (Liu et al. 2012). On the other hand, recent research has reported Mcr-dependent methanogenesis in *Archaeoglobaceae* from a hot spring (Buessecker et al. 2023). Based on this literature, acetate conversion to methane is possible through methanogenic pathways deduced from the reconstructed *Archaeoglobaceae* genomes (Buessecker et al. 2023), although *mcrA* gene sequences of *Archaeoglobaceae* were not detected in this study. Other potential candidates for methanogens or hydrocarbon-oxidizing archaea known to possess the Mcr or Acr (alkyl-S-CoM reductase) gene

in their genomes, such as *Ca*. Methanomethylicales, *Ca*. Nezhaarchaeales, *Ca*. Korarchaeia, *Ca*. Methanodesulfokores, *Ca*. Bathyarchaeota and *Ca*. Helarchaeales, were not detected in any of the samples.

### 3.7 Methanogen diversity in FW and culture samples based on the mcrA genes

A total of 29-46 clones per sample of the *mcrA* gene were grouped into 8 OTUs (Fig. 3b). Among these, 5 OTUs were also

detected in the methanogenic cultures of the FW samples (Table S4). Similar to the results of the 16S rRNA gene sequencing analysis, methanogen diversity changed with depth. *Methanobacterium* and *Methanocalculus* were dominant in the samples from depths above 1050 mbgs, whereas only thermophilic *Methanothermobacter* was recovered in the deeper samples at 1115 and 1373 mbgs, resulting in an overall predominance of hydrogenotrophic methanogens throughout the depths. In 559 and 754



mbgs, the methylotrophic methanogen *Methanolobus* was also detected as a major proportion. Acetoclastic methanogens, such
as *Methanosaeta* and *Methanosarcina*, were not detected in any of the samples.

These depth-related distributions of methanogens were roughly correlated with their growth temperature, as observed in
cultivation experiments (Fig. 3d and 3e, Table S4). Methane production from H2 and formate was observed at depths from 37
to 80 °C, except at 70 °C, whereas no methane production was observed in acetate-amended cultures from any FW samples
(data not shown). In cultures supplemented with methylated compounds, methane was detected only in 559 and 754 mbgs
samples, i.e., at 37 and 50 °C, respectively.

## 4 Discussion

This study examined how microbial diversity and processes are shaped by geothermal heat and associated geochemical
processes in deep subsurface environments, encompassing a range of temperatures from mesophilic to hyperthermophilic
conditions. The geochemical and hydrological analyses suggested the upward migration of volatile hydrocarbons derived from
lower oil deposits to upper aquifers, whereas no evidence indicated the mixing of water or less volatile n-alkanes between the
upper aquifers and lower oil deposits. Radiotracer measurements provided clear insight into the variations in potential
methanogenesis across different temperature regimes. Molecular and cultivation experiments indicate that these variations
align with the growth temperature ranges of major indigenous methanogens, except for hyperthermophilic ranges. Our previous
study also suggested that the upper temperature limit for growth in indigenous methanogens may reflect the potential for
methanogenesis in deep subseafloor sediments (Katayama et al., 2022). The correlation between the high potential for
acetoclastic methanogenesis and increased acetate concentration suggests that acetate generation, possibly driven by
geothermal heating (Parkes et al., 2007), stimulates acetoclastic methanogenesis in hyperthermophilic ranges. Although
syntrophic acetate oxidation (SAO) coupled with hydrogenotrophic methanogenesis is generally observed in high-temperature
environments (Conrad, 2023), the potential for this SAO methanogenesis is unlikely at hyperthermophilic depths at our study
site. This inference is drawn from the observation that the levels of $^{14}$C-activity from acetate were as much as two orders of
magnitude greater than those from carbonate in hyperthermophilic ranges. If SAO was the primary activity, the $^{14}$C activity
from acetoclastic methanogenesis would be much lower than that from hydrogenotrophic methanogenesis. This is because
$^{14}$C-CO2 produced from $^{14}$C-acetate via SAO is diluted into the biocarbonate pool in the incubated FW sample, leading to the
monitoring of $^{14}$C-CH4 production from CO2 with an extremely low level of $^{14}$C. We speculate that the predominant
*Archaeoglobaceae* may play a role in converting $^{14}$C-acetate to $^{14}$C-methane at greater depths (1115 and 1373 mbgs).

In addition to driving the acetoclastic methanogenesis observed in the radiotracer experiments, elevated concentrations of
acetate at greater depths (1115 and 1373 mbgs) may primarily serve for the dissimilatory reduction of sulfur compounds other
than sulfate, such as elemental sulfur and thiosulfate. This is inferred from the high relative abundance of sulfur-reducing,
rather than sulfate-reducing, microorganisms at those depths. This raises the question of how sulfur compounds are generated
to sustain these sulfur-reducing microbes, possibly reflecting the relatively high microbial population at greater depths. Based



on the sulfur isotopic analysis and previous literature, we suggest that elemental sulfur and/or thiosulfate are abiotically generated from sulfur compounds within crude oils according to the following reactions. Hydrogen sulfide can be generated through the thermal decomposition of sulfur compounds in crude oil (Zhu et al., 2017). The occurrence of this reaction is

supported by the lower $\delta^{34}S$ values of hydrogen sulfide in the FW sample from the lower oil deposits compared with those of sulfur compounds in oil [Fig. 2 (A)]. This generated hydrogen sulfide, together with other oil-derived volatile hydrocarbons, such as toluene and xylene, may migrate upward to upper aquifers and undergo chemical oxidation facilitated by the presence of Fe(III)-bearing minerals in the sediments. This process leads to the formation of elemental sulfur or thiosulfate. The occurrence of this reaction is supported by the lower $\delta^{34}S$ values observed for elemental sulfur than for hydrogen sulfide in the

FW sample from lower oil deposits [Fig. 2 (B)]. Previous studies have indicated the occurrence of chemical sulfide oxidation with metal oxides, such as FeOOH and magnetite ($Fe^{2+}Fe^{3+}2O_4$), in marine sediments (Holmkvist et al., 2011; Bottrell et al., 2008). Given that FeOOH is more readily reduced than magnetite during sediment burial, resulting in the preservation of magnetite (Canfield et al., 1992), magnetite likely serves as the primary agent responsible for sulfide oxidation at this study site. Despite the predominance of sulfur-reducing microorganisms in the upper aquifers as opposed to the absence of microbial

cells in the lower oil deposits, the differences in $\delta^{34}S$ values of hydrogen sulfide and elemental sulfur are not substantial between the upper and lower locations. This suggests that biological dissimilatory sulfur reduction [Fig. 2 (C)] is much less important than chemical sulfide oxidation [(B)] with respect to its influence on $\delta^{34}S$, which should be linked to differences in the sulfur flux of the biological and chemical processes in the upper aquifers.

A previous study indicated that the degree of generation of hydrogen sulfide from crude oil depends on various factors,

including the concentration of sulfur compounds in the oil, oil biodegradation coupled with microbial sulfate reduction and oil densification during burial (Zhu et al., 2017). At the study site, the concentration of sulfur compounds in oil-source rocks, as represented by the sulfur-to-carbon atomic ratio (S/C), was relatively low, ranging from 0.011-0.049, with an average of 0.025 (Suzuki et al., 1995). For comparison, high-sulfur oils exhibit an S/C ratio greater than 0.06 (Orr, 1986). This implies a low level of upward migration of hydrogen sulfide from oil deposits to upper aquifers. Nevertheless, our findings indicate that the

shifts in microbial population, diversity and function may be due to the chemical reactions induced by geothermal heating, which underscores the potential for a more dynamic biosphere–geosphere interaction that drives sulfur and carbon cycling in other deeply buried sedimentary environments.

*Data Availability.* DNA sequencing data are available at GenBank, as described in the Materials and Methods section. Other

datasets generated during the current study are available in the manuscript, supplementary information or from the corresponding author upon reasonable request.

*Competing Interests.* The authors have no relevant financial or nonfinancial interests to disclose.



*Author Contributions.* All the authors contributed to the study conception and design. Taiki Katayama, Hideyoshi Yoshioka and Yasuaki Hanamura contributed to the study conception and design and collected the samples. Hideyoshi Yoshioka, Kazuya Morimoto and Toshiro Yamanaka conducted water and sediment geochemical analyses. Taiki Katayama cultivated and analyzed the DNA. All the authors have read and approved the final manuscript.

*Acknowledgments.* We acknowledge Toshitaka Araki of the JX Nippon Oil and Gas Exploration Corporation for sample collection. We thank Chiwaka Miyako, Fumie Nozawa and Hanako Mochimaru for assistance in sample collection, sequencing analysis and cultivation experiments. We would like to thank Tatsuo Aono for his cooperation in the radiotracer experiments at the National Institute of Radiological Sciences (NIRS-QST). We would also like to thank Kazuya Morimoto for valuable comments regarding the chemical reactions of sulfur compounds and Takeshi Nakajima for managing this joint research project.

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



**Table 1.** Geochemical characteristics of the formation water samples.

| Depth (mbgs) | pH | ORP (mV) | Temp (°C) | $HCO_3^-$ (mg/l) | $Cl^-$ (mg/l) | $NO_3^-$ (mg/l) | $Br^-$ (mg/l) | $SO_4^{2-}$ (mg/l) | $PO_4^{3-}$ (mg/l) |
|---|---|---|---|---|---|---|---|---|---|
| 100 | 7.2 | -200 | 10.6 | 453 | 165 | 1.4 | 1.4 | 0.6 | bdl |
| 149 | 7.3 | -210 | 12.1 | 395 | 306 | bdl | 2.8 | 0.1 | bdl |
| 476 | 8 | -490 | 22.2 | 637 | 1750 | bdl | 17 | bdl | 1.5 |
| 613 | 7.6 | -290 | 25.6 | 2150 | 9500 | bdl | 96 | bdl | 6.4 |
| 715 | 7 | -380 | 28.3 | 2980 | 15600 | bdl | 170 | bdl | 7.5 |
| 943 | 8.1 | -450 | 35.4 | 3610 | 17100 | bdl | 140 | bdl | bdl |

| Depth (mbgs) | $Na^+$ (mg/l) | $NH_4^+$ (mg/l) | $K^+$ (mg/l) | $Mg^{2+}$ (mg/l) | $Ca^{2+}$ (mg/l) | $Fe^{2+}$ (mg/l) | DOC (mg/l) | Ace. (mg/l) | $\delta D$ (‰) |
|---|---|---|---|---|---|---|---|---|---|
| 100 | 146 | 4.3 | 22 | 38 | 37 | ndt | 7.5 | 0.2 | -79 |
| 149 | 160 | 15 | 20 | 66 | 50 | ndt | 5.2 | 0.029 | -76 |
| 476 | 1030 | 14 | 58 | 84 | 54 | 0.6 | 36 | 2.2 | -68 |
| 613 | 5640 | 79 | 230 | 270 | 89 | 1.4 | 85 | 0.63 | -46 |
| 715 | 9830 | 110 | 390 | 420 | 79 | ndt | 170 | 0.12 | -20 |
| 943 | 11100 | 210 | 440 | 310 | 100 | 1.6 | 220 | 16 | -12 |

Abbreviations: Ace., acetate; bdl, below the detection limit; ndt, not determined.