# Peer review of "The geothermal gradient shapes microbial diversity and processes in natural gas-bearing sedimentary aquifers"

_EGUsphere, 2024_

## Referee Comment (RC1)

**The geothermal gradient from mesophilic to thermophilic temperatures shapes microbial diversity and processes in natural gas bearing sedimentary aquifers**

**Comments:**

The study by Katayama et al. explored the microbial diversity and processes in sedimentary aquifers and sheds lights on effects of geothermal gradient on it. The topic of work is actual. Authors has estimated microbial range in sedimentary aquifers and is good for rational analysis of geochemical processes in subsurface areas driven by microbial activities. The manuscript is scientifically sound, and the methanogenic taxa results are obtained using both culture-based and amplicon-based techniques. Used methods are described in sufficient details, and obtained experimental data is appropriate. However, there is no mention of replications of the geochemical analysis of the samples.

The presented results are shown from sample analysis and laboratory experiments for methanogenesis. Results of metagenomics as Sequence Read Archives (SRA) are submitted in DDBJ under the BioProject accession number PRJDB16863. Novel interesting results are obtained concerning microbial diversity in those areas that adds to a viewpoint of distribution of microorganisms in deep subsurface environments.

Experimental procedures are adequately described, and literature is properly cited. Therefore, based on the details provided in the manuscript, I believe it has the potential to meet the standards.

Here are some pointers for the authors-

1) In the deepest sample (1373 mbgs), no mcrA gene could be quantified. However, if we consider the acetoclastic methanogenesis in that sample, it is the highest. Why? Discuss.
2) Line 138: Is it murA gene or mcrA gene?
3) Line 198: Remove 'of sulfur'

---

## Author Response (AR1)

**Responses to Referee #1**

*The study by Katayama et al. explored the microbial diversity and processes in sedimentary aquifers and sheds lights on effects of geothermal gradient on it. The topic of work is actual. Authors has estimated microbial range in sedimentary aquifers and is good for rational analysis of geochemical processes in subsurface areas driven by microbial activities. The manuscript is scientifically sound, and the methanogenic taxa results are obtained using both culture-based and amplicon-based techniques. Used methods are described in sufficient details, and obtained experimental data is appropriate. However, there is no mention of technical replications of the geochemical analysis of the samples.*

*The presented results are shown from sample analysis and laboratory experiments for methanogenesis. Results of metagenomics as Sequence Read Archives (SRA) are submitted in DDBJ under the BioProject accession number PRJDB16863. Novel interesting results are obtained concerning microbial diversity in those areas that adds to a viewpoint of distribution of microorganisms in deep subsurface environments.*

*Experimental procedures are adequately described, and literature is properly cited. Therefore, based on the details provided in the manuscript, I believe it has the potential to meet the standards.*

Thank you, Referee #1, for your thorough review and insightful comments on our study.

We appreciate your understanding of the importance of the research topic, acknowledgment of the scientific soundness of the manuscript, recognition of the appropriateness of the experimental procedures and data, and positive evaluation of the literature citation. We have also noted your point about the lack of mention of technical replications in the geochemical analysis. This is an important aspect to ensure the reliability and reproducibility of the results. Therefore, we have described this information in the revised manuscript as below, and also summarized it in the Table S3:

"The number of replication and the standard deviation value for each measurement are provided in Table S3." (page 5, lines 110-111)

"Each isotopic ratio was measured 6 times per sample, with the standard deviations of $\delta D$ and $\delta^{18}O$ less than 2.0‰ and 0.5‰, respectively." (page 5, lines 113-114)

"Each sulfur isotope ratio represents the average of duplicate measurements, with deviations less than 0.4‰." (page 5, lines 123-124)

*Q1. In the deepest sample (1373 mbgs), no mcrA gene could be quantified. However, if we consider the acetoclastic methanogenesis in that sample, it is the highest. Why? Discuss.*

Response: As you indicated, *mcrA* gene copy numbers could not be quantified despite the high potential activity of acetoclastic methanogenesis in the deepest sample. We believe that this is due to the limitation of the *mcrA* gene sequencing analysis because we did not obtain the mcrA gene sequences of *Archaeoglobaceae*, which may be involved in acetoclastic methanogenesis in the deepest sample. In the revised manuscript, we have addressed this limitation of the *mcrA* gene sequencing analysis as follows:

"Additionally, unlike the results from the 16S rRNA gene sequencing analysis, *Archaeoglobaceae* were not detected in the samples from 1115 and 1373 mbgs. The absence of *mcrA* gene sequences for *Archaeoglobaceae*, which might participate in acetoclastic methanogenesis in these samples (as described above), highlights the bias and limitation of PCR-based *mcrA* gene sequencing analysis. This issue has been documented by Vítězová et al. (2021), who indicated that *mcrA* gene analysis fails to detect specific methanogens, with their identification depending on the primers used. It could also explain the observed discrepancy between the highest $^{14}$C acetoclastic activity and the lowest mcrA gene copy numbers at these high-temperature depths." (page 11, lines 290-296)

*Q2. Line 138: Is it murA gene or mcrA gene?*

Response: It is *mcrA* gene. We have corrected it in the revised manuscript. (page 6, line 146)

*Q3. Line 198: Remove 'of sulfur'*

Response: We have modified the sentence accordingly. (page 8, line 218)

**Responses to Referee #2**

*The authors conducted both quantitative and qualitative molecular and geochemical analyses for the formation waters from a gas reservoir in Japan. The data set is descriptive but quite interesting. However, there are many issues that should be overcome prior to the publication.*

Thank you, Referee #2, for your insightful comments on our manuscript. We appreciate your interest in our work and acknowledge that there are areas requiring improvement. We are committed to addressing the issues identified to enhance the quality and clarity of our publication.

*Q1. The objective of this study is not clear, and the introduction section must be substantially reorganized.*

Response: As you suggested, we have thoroughly revised the introduction section (please also see the responses to the comments #2, 3, 4 & 5) as follows:

"The deep subsurface environment harbors a substantial fraction of Earth's prokaryotes (Mcmahon and Parnell, 2014; Magnabosco et al., 2018), constituting over 80% of the total prokaryotic biomass (Bar-On et al., 2018). The metabolic activities of these microorganisms play a pivotal role in global biogeochemical cycling, such as carbon, nitrogen and sulfur (Magnabosco et al., 2018; Aloisi et al., 2006). A prominent example is methane production. Much of the methane hydrate, the largest methane reservoir on Earth, is suggested to be generated by these subsurface microorganisms (Kvenvolden, 1995). Additionally, coalbed methane and shale gas reservoirs also may contain a significant amount of microbially-derived methane (Vinson et al., 2017).

In the absence of light energy, microorganisms inhibiting deep sedimentary environments rely on chemical energy derived from the oxidation of reduced substances in sediments with organic matter oxidation being particularly critical (Lovley and Chapelle, 1995; Jørgensen and Boetius, 2007; Arndt et al., 2013). The labile components are consumed during burial, limiting the availability of energy sources for microorganisms as sediment age increases (Middelburg, 1989). However, substantial populations of active microbial

cells have been observed even in deep buried sediments older than 16 million yr (Schippers et al., 2005). It has been hypothesized that temperature increase during burial stimulates thermal or biological degradation of recalcitrant organic matter, possibly sustaining microbial activities (Parkes et al., 2000). To address the fundamental questions of how microbial cells in the deep biosphere can survive with limited energy sources, this hypothesis has been examined through numerical simulations (Horsfield et al., 2006), laboratory incubation experiments (Parkes et al., 2007), and geochemical analysis (Malinverno and Martinez, 2015) of subseafloor sediments. Thus, while temperature increase during burial is posited to drive subsurface microbial activity, field observation-based microbiological research remains limited, and the mechanisms are not fully understood.

As sediment compaction progresses with burial, pore size and permeability decrease, reducing the living space, available water, and nutrients, thereby inhibiting microbial growth (Fredrickson et al., 1997). To investigate the impact of temperature increase on subsurface microbial ecosystem, it is essential to target subsurface environments where these inhibitory factors are minimized. Therefore, we focused on aquifers, which, even at great depths, maintain high porosity and permeability, providing ample living space and water for microorganisms (Fredrickson et al., 1997; Mcmahon and Chapelle, 1991; Lovley and Chapelle, 1995; Krumholz et al., 1997).

In this study, we targeted a gas field in central Japan, where aquifers exhibit a wide temperature range, spanning approximately 35 to 80 °C, due to a steep geothermal gradient (5 °C per 100 m) (Kato, 2018). We collected formation water (FW) from each aquifer and employed a comprehensive approach, including geochemical analysis, radiotracer measurements, molecular gene analysis and cultivation experiments to evaluate microbial diversity, community structure, and potential metabolic processes. Furthermore, in this field, high-temperature oil deposits situated deeper than the series of aquifers. We, therefore, also examined the impact of oil components on microorganisms in the upper aquifers. The aim of this study is to elucidate the effects of temperature rise and associated geochemical processes, such as the decomposition of sedimentary organic matter and petroleum formation, on microbial diversity, community structure, and metabolic processes, including the conversion of carbon and sulfur compounds." (pages 1-2, lines 28-61)

*Q2. L30-31: The first sentence does not link to any parts of this study.*

Response: The first sentence and related sentences in the introduction section has been modified so that it links to the aim of this study as follows:

"The deep subsurface environment harbors a substantial fraction of Earth's prokaryotes (Mcmahon and Parnell, 2014; Magnabosco et al., 2018), constituting over 80% of the total prokaryotic biomass (Bar-On et al., 2018). The metabolic activities of these microorganisms play a pivotal role in global biogeochemical cycling, such as carbon, nitrogen and sulfur (Magnabosco et al., 2018; Aloisi et al., 2006). A prominent example is methane production. Much of the methane hydrate, the largest methane reservoir on Earth, is suggested to be generated by these subsurface microorganisms (Kvenvolden, 1995). Additionally, coalbed methane and shale gas reservoirs also may contain a significant amount of microbially-derived methane (Vinson et al., 2017)." (pages 1-2, lines 28-34)

*Q3. L41-61: The paragraph of the site description should be moved to the result section. Most of the contents are not suitable for a part of Introduction section.*

Response: As you suggested, the site description has been moved to the Materials & Methods and Results sections in the revised manuscript as follows:

"The chemical and isotopic compositions of natural gases (Kaneko and Igari, 2020) indicate that methane dissolved in upper aquifers is primarily of microbial origin, whereas that from lower oil deposits primarily originates from oil-associated thermogenic processes (Fig. S1) based on the classification by Milkov and Etiope (Milkov and Etiope, 2018). In this gas field, gases are dissolved in FW and produced for commercial purposes by pumping gas-associated FW from upper aquifers. Crude oil and gases are also collected from lower oil deposits." (page 3, lines 73-77)

"The water temperatures ranged from 38 °C to 81°C in the FW samples from upper aquifers (Fig.1c) and from 67 to 96 °C in the samples from lower oil deposits (Table S1)." (page 7, lines 189-191)

*Q4. L50: The temperature boundary described in the previous study is not a rational why the authors did not analyze a sample of 90 ℃ or higher temperature in this study.*

Response: Within the upper aquifers where biogenic natural gas is deposited, there are no gas production wells with water temperatures above 81 °C. Therefore, we collected oil-associated formation water sample for microbial cell counts from lower oil deposits (1733 mbgs), in which water temperature was measured to be 96 °C. As a result, microbial cells were not observed in this sample, and molecular gene sequencing analysis was not conducted. Although these results were described in the original manuscript, we have modified the relevant sentences to clarify this point as follows:

"No microbial cells were observed in the oil-associated FW sample (1733 mbgs), in which water temperature was measured to be 96 °C (Table S3). Therefore, molecular gene sequencing analysis was not conducted on oil-associated FW samples from lower oil deposits." (page 10, lines 246-248)

We also note that, with the revision of the introduction section, references to the temperature boundary in both the main text and the title have been deleted.

*Q5. L63: It is not clear why the objective is important.*

Response: As described above, we have thoroughly revised the introduction section to clarify why the objective of this study is important. Our research investigates the potential metabolic activities of microorganisms in the deep subsurface environment, which play a significant role in global biogeochemical cycles. By focusing on aquifers with varying temperatures, this study aims to understand how microbial communities adapt and survive with limited energy sources, particularly in high-temperature environments. Additionally, we examine the impact of oil components on microbial activity, contributing to our knowledge of subsurface microbial ecology and its implications for methane production and other biogeochemical processes.

*Q6. The mcrA gene analysis was highly biased, and probably novel lineages of methanogen and anaerobic menthane oxidizers cannot be detected with the primer set. The authors should mention the limitation.*

Response: As you suggested, we have discussed the limitation of *mcrA* gene sequencing analysis in the revised manuscript as follows:

"The absence of *mcrA* gene sequences of *Archaeoglobaceae*, which can participate in acetoclastic methanoenesis in deeper samples (as described above), indicates the bias and limitation of the *mcrA* gene analysis, which may also explain the discrepancy between the highest 14C acetoclastic activity and the lowest *mcrA* gene copy numbers in the deeper samples at 1115 and 1373 mbgs."

>> We have modified this revision in response to your reply below (Q21) as follows:

"Additionally, unlike the results from the 16S rRNA gene sequencing analysis, *Archaeoglobaceae* were not detected in the samples from 1115 and 1373 mbgs. The absence of *mcrA* gene sequences for *Archaeoglobaceae*, which might participate in acetoclastic methanogenesis in these samples (as described above), highlights the bias and limitation of PCR-based *mcrA* gene sequencing analysis. This issue has been documented by Vítězová et al. (2021), who indicated that *mcrA* gene analysis fails to detect specific methanogens, with their identification depending on the primers used. It could also explain the observed discrepancy between the highest $^{14}$C acetoclastic activity and the lowest mcrA gene copy numbers at these high-temperature depths." (page 11, lines 290-296)

*<Minor comments>*

*Q7. L11: See above.*

Response: We also revised the abstract section in line with the changes in introduction section as follows:

"Deep subsurface microorganisms constitute over 80% of Earth's prokaryotic biomass and play an important role in global biogeochemical cycles. Geochemical processes driven by geothermal heating are key factors influencing their biomass and activities, yet their full breadth remains uncaptured." (page 1, lines 10-12)

*Q8. L15: "molecular gene analyses" instead of "molecular gene sequencing analyses"*

Response: We have modified the sentence accordingly. (page 1, line 14)

*Q9. L20, 26 and through the manuscript: Only 80˚C sample is a growth range of hyperthermophiles.*

Response: In the revised manuscript, we have avoided the use of terms indicating temperature range (i.e., mesophilic, thermophilic and hyperthermophilic). This is because exact temperature ranges for these terms vary between literatures, leading to potential misunderstandings.

*Q10. L39: microbial cell abundance*

Response: As a result of the significant revision of the Introduction section, the sentence containing this phrase has been removed.

*Q11. L44-46: Awkward sentence.*

Response: This sentence has also been removed due to the significant revision of the Introduction section.

*Q12. L119-: Please provide number of replicates for each.*

Response: To clarify the number of replicates for each, we have revised the relevant sentence as follows:

"The activity measurements were conducted in triplicate for each of the three cultivation periods and for each of the three radiotracers." (page 6, lines 130-131)

*Q13. L138: mcrA*

Response: We have modified the sentence accordingly. (page 6, line 146)

*Q14. L171: Please provide the gas pressure of N2/CO2.*

Response: The pressure of N2/CO2 was 0.1 MPa. We have added the information. (page 7, line 179)

*Q15. L176: enrichment instead of sample*

Response: We have modified the sentence accordingly. (page 7, line 184)

*Q16. L234-: Please define the nomenclature used in this study; e.g. GTDB, NCBI, SILVA, or ICNP with any others.*

Response: We used SILVA taxonomy because the taxonomic classification of 16S rRNA gene amplicon reads was performed based on the SILVA dataset in this study. (page 7, line 164)

*Q17. L248: In ICNP, Methanobacterota is effective but not Euryarchaeota.*

Response: Because the SILVA taxonomy still uses Euryarchaeota, we have modified the term as follows: "*Methanobacterota* (formerly *Euryarchaeota*)" (page 10, line 258)

*Q18. L255:  Deferribacter also uses iron or other electron acceptors that are likely available in the subsurface environments.*

Response: We agree with this point. We have emphasized the "common" catabolic metabolisms of abundant taxa detected in deeper samples. In addition, the chemical analysis of formation water (shown in Table S3) indicated that electron acceptors (iron, nitrate, nitrite, manganese) were almost depleted in the studied aquifers. Therefore, we consider it is not necessarily important to mention the potential of *Deferribacter* to use iron or other electron acceptors.

*Q19. L257-259: Too speculative.*

Response: As you suggested, we have removed the relevant sentences.

*Q20. L262: Nanobdellota instead of Ca. Nanoarchaeota.*

Response: Because the SILVA taxonomy still uses Nanoarchaeota, we have modified the term as follows: "Nanobdellota (formerly Ca. Nanoarchaeota)" (page 10, line 272)

*Q21. L268: Please clarify is this a result of a bias of methodology or not?*

Response: We believe that this is a result of primer bias. We have modified the relevant sentences as follows:

"The absence of mcrA gene sequences of Archaeoglobaceae, which can participate in acetoclastic methanoenesis in deeper samples (as described above), is due to the limitation in the mcrA analysis because the primers used did not match the mcrA gene sequences of Archaeoglobaceae. This limitation may also explain the discrepancy between the highest 14C acetoclastic activity and the lowest mcrA gene copy numbers in the deeper samples at 1115 and 1373 mbgs."

>> We have modified this revision in response to your reply below (*Q21*) as follows:

"Additionally, unlike the results from the 16S rRNA gene sequencing analysis, *Archaeoglobaceae* were not detected in the samples from 1115 and 1373 mbgs. The absence of *mcrA* gene sequences for *Archaeoglobaceae*, which might participate in acetoclastic methanogenesis in these samples (as described above), highlights the bias and limitation of PCR-based *mcrA* gene sequencing analysis. This issue has been documented by Vítězová et al. (2021), who indicated that *mcrA* gene analysis fails to detect specific methanogens, with their identification depending on the primers used. It could also explain the observed discrepancy between the highest $^{14}$C acetoclastic activity and the lowest mcrA gene copy numbers at these high-temperature depths." (page 11, lines 290-296)

*Q22. L282: observed in the enrichment cultures at*

Response: We have modified the sentence accordingly. (page 12, lines 298-299)

*Q23. L288-294: Appropriate tables and/or figures should be given for each sentence.*

Response: As you suggested, we added the appropriate tables and/or figures in parentheses at the end of each sentence. (pages 11-12, lines 304-312)

*Q24. L294: "ranges" should be deleted because no experiments at 90˚C in this study.*

Response: As described above, we have avoided the use of terms indicating temperature range.

*Q25. L339: Please provide accession numbers for mcrA gene sequences.*

Response: The accession numbers for *mcrA* gene sequences were described in page 6, line 163 in the original manuscript and page 7, line 171 in the revised manuscript.

<Continued minor comments (after authors replied in the interactive discussion)>

*Q1. "subsurface microbial ecosystem" instead of "subsurface microbial ecology"*

Response: We have modified the sentence accordingly. (page 2, line 49)

*Q4. I would ask the authors to provide information about water volume for each filter in the direct cell counts. Detection limit of the cell abundance in the M&M section is also helpful. "Thus" or "Therefore" instead of "Accordingly"*

Response: We have incorporated this information into the M&M section as follows: "Twenty milliliters of formation water sample was filtered through a 0.2-μm-pore-size Isopore membrane filter (Millipore), stained with SYBR Green solution (10 μg ml$^{-1}$) for 10 minutes, and observed under an epifluorescence microscope (BX51; Olympus, Tokyo, Japan). The detection limit for cell abundance was $4.3 \times 10^2$ cells ml$^{-1}$." (page 6, lines 139-141)

We have also changed "Accordingly" to "Therefore". (page 10, line 247)

*Q6: "might" instead of "can". Delete "deeper"*

Response: As you suggested, we have modified the phrases from "which can participate in acetoclastic methanoenesis" to "which might participate in acetoclastic methanoenesis" and from "*mcrA* gene copy numbers in the deeper samples at 1115 and 1373 mbgs" to "*mcrA* gene copy numbers at these high-temperature depths". (pages 11, lines 292-296)

*Q16: The response is not clear whether it was declared in the revised manuscript or just mentioned in this letter.*

Response: The revision was incorporated into manuscript. Specifically, we included the sentence "In this study, the nomenclature of prokaryotic lineages was based on this SILVA taxonomy." on page 7, line 164 of the revised manuscript.

*Q18: In my understanding, potential electron donors and/or acceptors are sometimes depleted in closed subsurface biosphere. I do not agree with the logic of the authors. The authors likely pick up subjective information based only on rRNA gene sequences.*

Response: Our suggestion regarding the potential for dissimilatory reduction of sulfur compounds in deeper samples was supported not only by rRNA gene sequencing but also by sulfur isotopic analyses. However, we acknowledge your concern about the potential subjectivity of information based solely on rRNA gene sequences. In response, we have included further details of the potential metabolisms of dominant taxa in the Results section of the revised manuscript as follows:

"In the samples deeper than 1000 mbgs, a single ASV accounted for ≥50% of the total sequences in each sample. These genes were identified as closely related to *Deferribacter desulfuricans* (100% sequence similarity, in the 1049 mbgs sample), *Thermacetogenium phaeum* (93% sequence similarity, in the 1115 mbgs sample) and *Thermanaeromonas toyohensis* (97% sequence similarity, in the 1373 mbgs sample). These species can commonly utilize sulfur compounds, such as thiosulfate and elemental sulfur, as electron acceptors and acetate as an electron donor (Takai et al., 2003; Hattori et al., 2000; Mori et al., 2002). Additionally, these genera are known to utilize a variety of electron acceptors: *Deferribacter* can use nitrate, iron (III), or manganese (Slobodkina et al., 2009); *Thermacetogenium* can use sulfate (Hattori et al., 2000); and *Thermanaeromonas* can use nitrate, nitrite, sulfate, or fumarate (Gam et al., 2016)." (pages 10, lines 262-269)

*Q20: The response is not consistent with the case of Euryarchaeota.*

Response: We have updated the manuscript to reflect the change from *Euryarchaeota* to *Methanobacteriota* (formerly *Euryarchaeota*). This modification is present on page 10, lines 258 and 270, as well as in Fig. 3.

*Q21: Did the authors confirm this in silico? If not, appropriate reference should be listed.*

Response: No, we did not confirm this *in silico*. We inferred the limitation of *mcrA* gene sequencing analysis because we did not obtain the *mcrA* gene sequences of *Archaeoglobaceae*, despite their abundance in the 16S rRNA gene sequences and the high

potential activity of acetoclastic methanogenesis observed in the deep samples in the $^{14}$C-tracer experiments.

In the revised manuscript, we have addressed this limitation of the *mcrA* gene sequencing analysis with supporting the literature by Vítězová et al. 2021 as follows:

"Additionally, unlike the results from the 16S rRNA gene sequencing analysis, *Archaeoglobaceae* were not detected in the samples from 1115 and 1373 mbgs. The absence of *mcrA* gene sequences for *Archaeoglobaceae*, which might participate in acetoclastic methanogenesis in these samples (as described above), highlights the bias and limitation of PCR-based *mcrA* gene sequencing analysis. This issue has been documented by Vítězová et al. (2021), who indicated that *mcrA* gene analysis fails to detect specific methanogens, with their identification depending on the primers used. It could also explain the observed discrepancy between the highest $^{14}$C acetoclastic activity and the lowest mcrA gene copy numbers at these high-temperature depths." (page 11, lines 290-296)